# The Use of Digital Resources in Teaching during the Pandemic: What Type of Learning Have They Promoted?

**Beatriz Cabellos \*** , **M. Puy Pérez Echeverría and Juan Ignacio Pozo**

Department of Basic Psychology, Faculty of Psychology, Autonomous University of Madrid, 28049 Madrid, Spain
\* Correspondence: beatriz.cabellos@uam.es

**Abstract:** The COVID-19 pandemic induced an accelerated transition to digital teaching in all countries. We ask ourselves whether this massive use of digital resources promoted student–centred, dialogical, and multimodal teaching, as shown in some experimental studies, or whether, on the contrary, these resources were used only as a substitute for the teacher's voice, maintaining content-centred teaching. We analysed 269 activities carried out during the lockdown by teachers in Spain through the System of Analysis of Teaching Activities. This analysis system considered the resources used and the learning they promoted within activities. In general, the activities were content-centred independently of the resource used. However, in a few cases, activities were student–centred. The greatest systematic difference between the two types of activities was related to who managed these resources. Only when the student managed the digital resources were the activities student-centred. Conversely, when the tasks were content-centred, teachers managed the resources. These results indicate the need to achieve proven ICT integration in education, which in turn requires a boost in teacher training aimed at both familiarising teachers with the use of digital resources and, above all, promoting a change in teachers' conceptions about their use.

**Keywords:** digital resources; digital teaching; ICT; elementary and secondary education; information processing; teaching/learning strategies; COVID-19





## 1. Introduction

### 1.1. Digital Literacy in 21st-Century Education

The development of both digital competences and general competences related to the treatment of information are among the most important challenges for education in the 21st century. Both challenges are related. Several authors suggest that the educational integration of Information and Communication Technology (ICT) could contribute to the development of some 21st-century skills [1]. These technologies could facilitate student-centred teaching [2,3] by allowing content and learning pace to be adapted to each student and, therefore, attain greater achievement of these competences [4]. Moreover, ICT's multimodal nature [5–7] would contribute both to the possibility of transcending the traditional verbal content that usually dominates the curriculum [8,9] and to the adoption of different perspectives. This plurality of voices and languages turns ICT into potentially dialogical environments based on interaction and collaboration [4], as opposed to the monologic, teacher-centred nature of traditional education, fostering more complex and effective learning [10].

Extensive experimental research shows that the use of digital resources based on these principles has a generally positive effect on learning in different subject areas and ages, compared to traditional educational interventions [11]. For example, the use of touchscreens significantly enhances learning in young children [12]. The educational use of video games produces moderate but consistent effects on learning in different curriculum areas [13,14], including some commercial video games originally designed for entertainment [15].

However, in contrast to these studies showing ICT's educational benefits, abundant research exists demonstrating no or negative effects on learning [16]. These results are particularly marked when we compare controlled experimental research with large-scale national or international studies that measure the effect of ICT use on learning in "natural" classroom contexts, such as the OECD's PISA studies [17]. Indeed, after analysing data from these studies, Biagi and Loi [18] found that results improved when students engaged in more ICT-mediated activities outside the classroom but that they decreased the more ICT was used in the classroom for school purposes. Bluntly speaking, according to these studies, the more ICT is used in the classroom, the less learning takes place. Consequently, in its report, the OECD [17] concludes that "the results also show no appreciable improvements in student achievement in reading, mathematics or science in the countries that had invested heavily in ICT for education".

What is the reason for the discrepancy between controlled research, conducted in experimental laboratories, and these large-scale studies? Many factors could explain this gap, but one difference is that these experimental situations have been carefully designed based on the principles mentioned (student-centred approach, use of multiple codes and languages, and fostering dialogue and collaborative learning), whereas the usual large-scale classroom work reported in these international studies is mediated by the activity of teachers who often have little training in the use of ICT [19].

On the other hand, other works unveil the most common uses of classroom digital technologies as searching for information on the internet or watching videos [20–22]. These studies indicate that teachers use ICT to present content, predominantly in textual code, which students must acquire with little or no transformation [21,23]. For their part, students usually produce answers or essays based on cut-and-paste [24]. In these activities, the student takes a passive role both in managing the digital resource used and in carrying out the activity. It is also observed that, unlike occurrences in non-formal and informal education environments [25], the use of social networks, forums, and blogs is infrequent, and they are used in unidirectional communications, with little dialogue [21], like a messaging service [20]. Therefore, there is no integration of different perspectives that could lead to shared knowledge.

In short, although the dominant theoretical models indicate the need to promote new, student-centred, multimodal, and dialogical uses of ICT that favour digital literacy [4], it is unclear whether these technologies have usually been aimed at this learning or more oriented towards supporting or replacing the teacher's voice in more traditional, content-centred educational scenarios. In these circumstances, the forced use of digital technologies because of the COVID-19 pandemic has been a privileged space to analyse the uses that teachers make of ICT and to study whether these are supported by the principles that promote better learning.

*1.2. Digital Literacy during the Pandemic. What Happened in Schools during Lockdown?*

The forced use of digital technologies in classrooms due to the COVID-19 pandemic can be understood as a global critical incident [26,27] that has altered the usual practices of teachers worldwide. This need for ICT-based distance learning meant that teachers had to make a great effort to adapt, increasing their workload considerably [28]. However, our interest lies not in analysing this effort, which should be recognised and praised, but in studying the activities they proposed to their students. According to the literature, critical incidents can be solved either by rethinking those practices or by seeking the safety of entrenching deeply held beliefs about how to teach in the classroom. In this way, the pandemic is an ideal context in which to analyse the uses that have been made of ICT.

Many of the studies carried out during this period focused on studying which resources have been used most in teaching [29,30]. The use of resources such as YouTube and downloadable text files stood out. There has also been an increase in the use of platforms that allow students to complete exercises from which they receive automatic feedback, facilitating teaching assessment (for instance, platforms from different editorials such as

Pearson and McGraw Hill). Confinement also seems to have encouraged the use of both asynchronous communication platforms, via Classroom, WhatsApp or Telegram, and synchronous communication platforms, such as Teams and Zoom. However, these papers do not elaborate on the activities that have been promoted through these resources.

In this context, our main objective was to identify the activities carried out with these ICT resources in primary schools (6–12 years old) and secondary schools (12–18 years old). Can we expect that those features that optimise learning in digital contexts (student-centred, multimodal, and dialogical) were prioritised or that more traditional educational practices (content-centred, single-code based, and monologic) were reproduced? In other words, we ask whether this ICT-based teaching has been student-centred and has sought the construction of knowledge and competences or whether it promoted the acquisition of specific content in a more reproductive way. We can expect student-centred teaching to focus on open-ended activities in which the learner manages resources [4]. These types of activities encourage both the acquisition of knowledge and the learning of the strategies necessary to search for, interpret, analyse, organise, or communicate information and thus to appropriate knowledge and make sense of it [31]. Assessment would be part of the learning process itself and would have not only an accrediting function but also a formative one [32]. In contrast, content-centred teaching promotes closed activities, in which students are limited to accessing information provided and managed by teachers [1,33], encouraging only the acquisition of reproductive knowledge and not its transformation. Assessment is made based on the product of activity carried out by the students.

Finally, we were interested in finding out whether any of the teachers' characteristics were related to the way they used digital resources. The literature on this subject presents very inconclusive results that vary from study to study and that in general are more aimed at uncovering attitudes towards the use of ICT in education than at analysing real differences in its use. For example, there are studies [33] that have found gender differences in these attitudes, whilst others have not [34,35]. Other studies have found an inverse relationship between teachers' age or years of teaching experience and how digital resources are accepted [36,37], while others do not confirm these differences [34,35]. The results on the influence of the educational stage at which classes are taught seem to be more conclusive, showing that secondary school teachers seem to be more likely to use digital resources than primary school teachers. In addition, prior use of ICT in the classroom seems to predict its greater subsequent use.

Therefore, in this paper we first set out (objective 1) to investigate which digital resources were used in primary and secondary education teaching during the pandemic, comparing the frequency of their use. Beyond this, we also wished to investigate what the resources were used for, i.e., (objective 2) to identify the types of learning teachers carried out during the confinement. From there, we were interested (objective 3) in determining the specific use made of each digital resource found. Finally, cross-sectionally to these objectives, we wished (objective 4) to analyse the influence of variables associated with teachers in these different uses of digital resources.

## 2. Materials and Methods

### 2.1. Task and Procedure

As part of a broader investigation [33,38] carried out during the months of lockdown, Spanish primary (6–12 years old) and secondary (12–18 years old) school teachers were asked to describe an activity they had carried out with their pupils, indicating its objectives, the roles of the pupils and teacher, assessment methods, resources needed, and difficulties encountered. In this work, we focused on the different uses made of ICT resources. The task was implemented in the Qualtrics tool and sent to teachers all over Spain. The data was collected from 4 May to 13 June 2020, during the school lockdown. To encourage participation, a prize draw of EUR 75 worth of educational material was held among the participants.

## 2.2. Participants

A total of 287 teachers sent a description of an activity. Six of these teachers submitted two activities and one submitted three. We eliminated from the sample the activities of 26 teachers who did not provide sufficient information for the analysis. We therefore obtained a sample of 269 activities described by 261 teachers. The personal and professional variables of the participants (gender, years of experience as teachers, stage and speciality they taught, type of centre, and previous experience with ICT) were also collected. The information about these variables can be consulted in Appendix A Table A1.

## 2.3. Design of a Category System for the Analysis of Practices

To carry out the study we used an ex post facto retrospective design in which the dependent variables were the teachers' responses, analysed using a system of categories System of Analysis of Teaching Activities (SATA), adapted from the System of Analysis of Practices of Instruction and Learning (SAPIL) [39,40], which had been used in previous work on the observation of teaching practices [38] (see Tables 1 and 2).

**Table 1.** SATA categories: type of resource and management.

| Resource | Categories of Analysis | | Definition |
|---|---|---|---|
| | **Audiovisual** | | **Graphic, Image, Sound, and/or Video Resources.** |
| Type of resource | Text files | Informative | Texts aimed at providing data and information. |
| | | Questionnaire-based | Texts which specify questions to be answered by the students. |
| | | Expository | Texts providing theories and arguments. |
| | | Expressive | Narrative or poetic literary texts. |
| | | Multiple | Texts which present a combination of two or more codes (multimodality). |
| | Communication platforms | Social media platforms | Resources which promote communication of a broad group or the whole classroom community. |
| | | Email | Resources aimed at two-directional, written, asynchronous communication and sending of files. |
| | | Video call | Communication by video and audio in synchronised form. |
| | Software | | Programs aimed at carrying out different learning activities (Kahoot, Genially, Scratch, etc.). |
| Resource management | Teacher | Selection | The teacher selects the resource. |
| | | Production | The teacher creates the resource. |
| | Student | Selection | The student selects and manages the resource. |
| | | Production | The student creates and manages the resource. |

To refine the design of SATA, inter-judge analysis was carried out both for the final definition of the categories and for the inclusion of the activities within them. This analysis was carried out in several phases. During the first phase, the judges worked on the selection of categories and the examination of some activities. Once an initial analysis system was in place, during the second phase, two groups of three judges, collaborators of the research team, analysed 50 activities (17.40% of the total) chosen randomly, obtaining a high inter-judge agreement (Kappa indices between 0.78 and 0.86). In this phase, categories that were not used in the activities described by the teachers were eliminated and the criteria for definition and inclusion in the different categories were adjusted. After this analysis, the remaining activities were randomly divided among three of the six researchers involved in the previous phase, who independently analysed the different activities. Disparities in the inclusion of categories were resolved through discussion and final consensus.

**Table 2.** SATA categories: characteristics of the activity.

| Characteristics of the Activity | Categories of Analysis | Definition |
|---|---|---|
| | **Audiovisual** | **Graphic, Image, Sound, and/or Video Resources.** |
| Learning outcomes | Verbal learning | Learning of facts and concepts. |
| | Procedural learning | Learning of techniques and strategies. |
| | Attitudinal learning | Learning of attitudes and values. |
| Learning processes | Reproductive learning | Literal learning (facts, techniques, and attitudes). |
| | Constructive learning | Learning that requires the transformation of prior knowledge into new meanings (concepts, strategies, and values). |
| Information processing | Information acquisition | Surface level processing, where information is processed in much the same way as it was received. |
| | Information interpretation | Translation of the information received into a different code or format. |
| | Information analysis | Explicit inferences beyond the information provided, drawing new conclusions, etc., not explicitly included in it. |
| | Organisation of information | Explicitly relating one set of information or knowledge to another, generating meaningful, causal, or hierarchical relationships between them. |
| | Communication of information | Explicit communication of acquired knowledge. |
| Information management | Access to the information | The student accepts the information proposed by the teacher. |
| | Search for information | The student actively looks for information. |
| Treatment of the information | Reproduce information | Substantial repetition of the information. |
| | Produce information | Creation of new content from the information. |
| Type of task | Closed task | Tasks with well-defined procedures where the answers may be classed as correct or incorrect. |
| | Open task | Tasks where the resolution procedures must be defined and where several responses are possible. |
| Evaluation | Evaluation of the product (summative) | Final and qualifying evaluations. |
| | Evaluation of the process (formative) | Evaluations of the procedure, aimed at thinking about the process itself. |
| Social organisation | Individual task | Activities which the student has to complete on their own. |
| | Class group task | Activities undertaken by the whole group and the teacher. |
| | Small group work | Activities in small groups of students. |

The finally developed system, as can be seen in Table 1, collected the resources used by teachers in their activities. These resources were categorised as text files, audiovisual resources, communication platforms, and software. Text files were subdivided into informative, expository, expressive, multiple, and questionnaire-based. Audiovisual resources consisted mainly of videos, although audio and images were also included in this categorisation. However, the low frequency of occurrence of these last two subcategories led us not to consider them in the analyses. Communication platforms consisted of social media platforms, email, and video calls. Finally, the software category included only those programs used to carry out learning activities. This system also differentiated whether these resources were managed by the teacher or the students and whether this management consisted of the selection of products built by others or the development of the resource itself. This analysis did not apply in the case of communication platforms which, by definition, are interactive and where it was not possible to establish these differences.

As can be seen in Table 2, SATA also allowed a distinction to be made between the possible types of activity, considering the outcomes and processes of learning, the processing, management, and treatment of information, the type of task, the mode of

assessment, and the social organisation involved. The learning outcomes refer to what is learned through the activity, while the learning processes would be the types of learning more oriented to the reproduction of content or to the construction of knowledge that mediate those learnings. The type of information processing that students had to do was also taken into account [39]. In "information management" we considered whether the students were limited to receiving information provided by their teacher or whether they had to search for it autonomously. Also considered was whether the task only required reproducing that information or whether it was necessary for the student to generate new content. As for the modes of assessment, a distinction was made between those assessments oriented towards training, focused on processes, and helping the student to improve their performance, and those activities more oriented towards the grading of the final product. Finally, the social organisation of the activity was considered. Examples of these categories can be found in Appendix A (Tables A2–A5).

*2.4. Data Analysis*

Each activity was analysed in terms of these categories. We counted the frequencies with which the digital resources were used and the different characteristics of the activities appeared (see Tables 1 and 2). McNemar analyses were carried out to identify differences between the frequency of appearance of the different digital resources used (objective 1). Again, McNemar analyses were carried out to identify differences between the frequency of the different characteristics of the activities (objective 2). We also checked whether there were relationships between the frequency of occurrence of these activity categories and the resources used (objective 3).

We took into account whether the activities were similar or, on the contrary, differed depending on the type of digital resource used. For this analysis, we used the statistic chi-square ($\chi^2$) and Adjusted Standardised Residuals (ASR).

Finally, we tested whether resource use could be related to teaching characteristics (objective 4), again using the $\chi^2$ statistic and ASR. However, in those analyses where more than 25% of the boxes had a frequency of less than 5, the Fisher statistic was used instead of $\chi^2$.

**3. Results**

The activities described by the teachers showed the presence of different types of digital resources, whose frequency also varied (see Figure 1) (objective 1). The most frequently named resources were audiovisuals and text files, which were used in more than half of the activities, while software programs were only employed in just over 16% of activities. Significant differences were found between the use of audiovisual and text files versus communication platforms, as well as between these and software programs ($p < 0.001$).

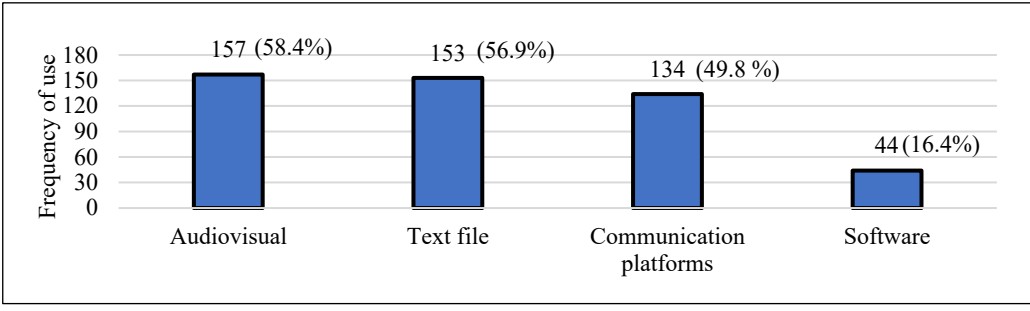

**Figure 1.** Frequency of use of the different digital resources. Note: The percentages are calculated over the total number of activities (N = 269). More than one resource could be used in the same activity, so the sum of them is greater than 100%.

Therefore, concerning the first objective, we found that the most employed resources were audiovisuals and text files for the presentation of content or the completion of tasks. But for what purpose were these digital resources used, and what learning were they intended to promote? Concerning this second objective, as can be seen in Table 3, the activities described by the teachers were directed towards reproductive learning ($p < 0.001$) and in which verbal outcomes predominated compared to procedural and attitudinal ones ($p < 0.001$). These teaching activities primarily required students to acquire or incorporate information without elaborating on it. The least frequent demand was related to organising or analysing this information to extract new knowledge from it.

**Table 3.** Frequency of the type of activity.

| Dimension | Category | n | % |
|---|---|---|---|
| Learning outcomes | Verbal learning | 185 | 68.8 |
| | Procedural learning | 114 | 42.4 |
| | Attitudinal learning | 81 | 30.1 |
| Learning processes | Reproductive learning | 251 | 93.3 |
| | Constructive learning | 71 | 26.4 |
| Information processing | Information acquisition | 213 | 79.2 |
| | Interpretation of information | 88 | 32.7 |
| | Analysing information | 26 | 9.7 |
| | Organising information | 51 | 19.0 |
| | Information communication | 103 | 38.3 |
| Information management | Accessing information | 173 | 64.3 |
| | Searching for information | 75 | 27.9 |
| Treatment of the information | Reproducing information | 154 | 57.2 |
| | Producing information | 130 | 48.3 |
| Type of task | Closed | 183 | 68.0 |
| | Open | 103 | 38.3 |
| Evaluation | Product evaluation (summative) | 215 | 79.9 |
| | Process evaluation (formative) | 35 | 13.0 |
| Social organisation | Individual | 235 | 87.4 |
| | Class group | 59 | 21.9 |
| | Small groups | 13 | 4.8 |

Note: The percentages are calculated over the total number of activities (N = 269). More than one resource could be used in the same activity, so the sum of them is greater than 100%.

All differences were significant ($p < 0.01$), except for the relationship between interpretation and communication, where no differences appeared. Thus, we can say that, in general, the activities required rather superficial information processing from students. Moreover, these results are consistent with what teachers explained that students should do in these tasks. In general, students were expected to access materials provided by teachers (64.3% of the activities) rather than search for information (27.9% of the activities) ($p < 0.001$). There was also a tendency to propose activities aimed at reproducing information (57.2%) rather than asking students to generate new content (48.3%), but these differences were not significant. Accordingly, more closed than open activities were carried out ($p < 0.001$) and assessment was more focused on the product than the process ($p < 0.001$). Finally, the most common activities were individual, followed by the class group, with hardly any presence of student group work, which could encourage interaction or collaboration between students ($p < 0.001$).

In short, activities were predominantly oriented towards the acquisition of verbal content and promoted reproductive processes. These activities were characterised by being closed and individual, in which the learner focused on accessing the information provided by the teacher and in which the final product was assessed, rather than the process used to achieve that result. They were therefore content-focused activities managed specifically by the teacher. A typical example of this type of activity was reported by a teacher of 6–9-year-old children who asked her students to watch a video story and answer a series of questions (example 21 in Appendix A Table A4) or that reported by a teacher of 9–12-year-old children who provided her students with notes which they had to study to complete a knowledge acquisition test in a self-correcting format (example 38 in Appendix A Table A5).

This general portrait is enriched by analysing how each digital resource was specifically used in the classroom. To this end, in keeping with our third objective, we focused on analysing which characteristics of the activity were most frequent depending on the resource used and whether the type of activity changed depending on who managed the resource and how. Therefore, we begin with the most frequently used resources, which, as we have seen, were audiovisual. As can be seen in Table 4, the chi-square and residual analyses showed that teachers sent their students audiovisual files so that they could access the information contained in them ($p < 0.001$, ASR = 3.1). These files were not intended to be used to organise information ($p < 0.05$, ASR = $-2.1$). However, when it was the students who produced such resources, there were more activities aimed precisely at this organisation ($p < 0.05$, ASR = 2.4), while those that only accessed them were less frequent ($p < 0.001$, ASR = $-3.5$). An example of this type of activity was the case of a teacher who asked her 12–16-year-old students to create a story from the covers of different books, reorganising the information they provided to create a new story (see example 12 in Appendix A Table A3). However, when it was the teacher who selected ($p < 0.01$, ASR = 3.1) or produced the resource ($p < 0.01$, ASR = 2.6), the activities only required access to the information, as can be seen in the following example of a teacher of 9–12-year-old students who provided links to self-created audiovisual montages, which the students had to review and then answer a series of questions on (example 10 in Appendix A Table A3).

**Table 4.** Relationship between audiovisual resource use and its management: who manages it and how.

| Category | | Relationship between Categories | Relationship | Statistical Data |
|---|---|---|---|---|
| Audiovisuals (total) | | Access to the information | + | $p < 0.001$, ASR = 3.1 |
| | | Organisation of information | − | $p < 0.05$, ASR = $-2.1$ |
| Who manages the text file information? | The teacher selects | Access to the information | + | $p < 0.01$, ASR = 3.1 |
| | Teacher produces | Access to the information | + | $p < 0.01$, ASR = 2.6 |
| | Student produces | Access to the information | − | $p < 0.001$, ASR = $-3.5$ |
| | | Organisation of information | + | $p < 0.05$, ASR = 2.4 |

In the case of text files, as anticipated in the method, different types were distinguished according to their communicative function (questionnaire-based, informative, expository, expressive, informative) (see Figure 2). The most frequently used text files were multiple texts, which were significantly more frequent than expressive or expository texts ($p < 0.05$). These resources were used in individual activities ($p < 0.01$, ASR = 2.7) (see Table 5) aimed at verbal learning ($p < 0.01$, ASR = 2.9), especially when it came to questionnaire-based communication ($p < 0.05$, ASR = 2.3), although this was not the case with literary-expressive texts ($p < 0.05$, ASR = $-2$). As in the case of audiovisuals, students were limited to accessing the information ($p < 0.05$, ASR = 2.2), both when teachers selected the texts ($p < 0.001$, ASR = 4.4) and when they produced them themselves ($p < 0.01$, ASR = 3.1). This result can be seen in the following activity of a teacher of 9–12-year-old children. This teacher gave her students notes with different links for them to access (example 27 in Appendix A

Table A5). Nevertheless, when students selected the texts ($p < 0.001$, ASR = −3.5) or wrote them themselves ($p < 0.01$, ASR = −2.6), they did not simply access the information. For example, the following teacher of 12–16-year-old students (example 39 in Appendix A Table A5) stated that she wanted her students to be able to write a piece of text of no more than 100 words in which they recount an encounter with a certain character with appropriateness, coherence, and cohesion [ . . . ]. In short, this teacher provided her students with an evaluation rubric to help them better control their activity. This way of evaluating is an example of a trend that our data shows. The evaluation is more formative when the texts are created and managed by the students instead of created and managed by the teacher ($p < 0.05$, ASR = 2.2).

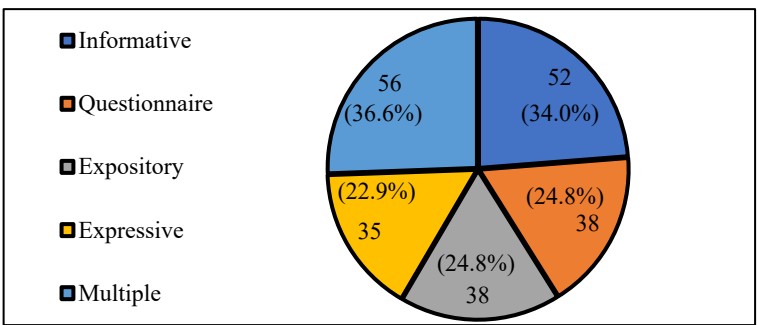

**Figure 2.** Frequency of usage of the different text files. Note: The percentages are calculated over the total number of activities which use text files (N = 153). More than one resource could be used in the same activity, so the sum of them is greater than 100%.

**Table 5.** Relationship between test file resource use and its management: who manages it and how.

| Category | | Relationship between Categories | Relationship | Statistical Data |
|---|---|---|---|---|
| Text files (total) | | Verbal learning | + | $p < 0.01$, ASR = 2.9 |
| | | Access to the information | + | $p < 0.05$, ASR = 2.2 |
| | | Individual task | + | $p < 0.01$, ASR = 2.7 |
| | | Process evaluation (formative) | + | $p < 0.05$, ASR = 2.2 |
| Type of text files | Questionnaire-based | Verbal learning | + | $p < 0.01$, ASR = 3.1 |
| | | Access to the information | + | $p < 0.01$, ASR = 3.0 |
| | | Process evaluation (formative) | − | $p < 0.05$, ASR = −2.2 |
| | Expressive text | Verbal learning | − | $p < 0.05$, ASR = −2.0 |
| Who manages the text file information? | Teacher selects | Access to the information | + | $p < 0.001$, ASR = 4.4 |
| | Teacher produces | Verbal learning | + | $p < 0.001$, ASR = 3.7 |
| | | Access to the information | + | $p < 0.01$, ASR = 3.1 |
| | Student selects | Process evaluation (formative) | − | $p < 0.001$, ASR = −3.5 |
| | Student produces | Access to the information | − | $p < 0.01$, ASR = −2.6 |
| | | Process evaluation (formative) | + | $p < 0.01$, ASR = 2.8 |

There were differences in the frequency with which different communication platforms were used (see Figure 3), with video calls being the most frequent, followed by social media platforms and finally email. All differences were significant ($p < 0.05$). These resources (Table 6) were mainly used for students to search for information ($p < 0.05$, ASR = 2.3) which they then had to reproduce ($p < 0.05$, ASR = 2.0). It is also interesting that they were not employed in individual activities ($p < 0.001$, ASR = −4.4) but when the whole classroom was involved ($p < 0.001$, ASR = 6.4). Video calls, in addition to following this same pattern, were characterised by promoting verbal learning ($p < 0.05$, ASR = 2.3). In turn, emails were used in activities that promoted active information seeking ($p < 0.001$,

ASR = 3.9). However, social media platforms were used more frequently for students to access a given information ($p < 0.05$, ASR = 2.3), which seems to indicate that the teacher is the one who managed them for sharing resources with their students rather than as a space for more horizontal interaction. The following activity of a secondary teacher is an example of this type of educational practice: "I develop my English classes by sharing my materials on the subject's blog, where the materials are posted, both written texts and audiovisual material" (example 6 in Appendix A Table A2).

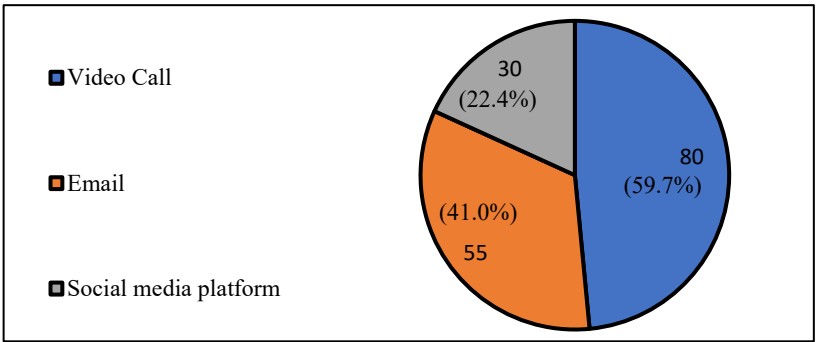

**Figure 3.** Frequency of usage of the different communication platforms. Note: The percentages are calculated over the total number of activities which use communication platforms (N = 134). More than one resource could be used in the same activity, so the sum of them is greater than 100%.

**Table 6.** Relationship between communication platforms and activity promoted.

| Category | | Relationship between Categories | Relationship | Statistical Data |
|---|---|---|---|---|
| Communication platforms (total) | | Search for the information | + | $p < 0.05$, ASR = 2.3 |
| | | Reproduce information | + | $p < 0.05$, ASR = 2.0 |
| | | Individual task | − | $p < 0.001$, ASR = −4.4 |
| | | Class group task | + | $p < 0.001$, ASR = 6.4 |
| Types of communication platforms | Video calls | Verbal learning | + | $p < 0.05$, ASR = 2.3 |
| | | Individual task | − | $p < 0.001$, REC = −6.0 |
| | | Class group task | + | $p < 0.001$, ASR = 8.2 |
| | Emails | Search for the information | + | $p < 0.001$, ASR = 3.9 |
| | Social media platforms | Access to the information | + | $p < 0.05$, ASR = 2.3 |

Finally, educational software programs (16.4% of the activities) were employed, as shown in Table 7, in closed activities ($p < 0.05$, ASR = 2.1), which again required the student to access the information ($p < 0.05$, ASR = 2.3) and then reproduce it ($p < 0.05$, ASR = 2.3). However, they were frequently also used in formative assessment-oriented activities ($p < 0.05$, ASR = 2.1). This seemingly contradictory result may occur because this type of assessment was most frequent when students produced their own software. This software was not used for attitudinal learning ($p < 0.01$, ASR = −3), to favour communication ($p < 0.2$, ASR = −2.3), or to produce new content ($p < 0.001$, ASR = −3.7). Along the same lines, a negative relationship can be observed between the production of materials using software by the teacher and open activities ($p < 0.05$, ASR = −2.3) in which the student must produce new content ($p < 0.05$, ASR = −2.7) and which are evaluated in a formative way ($p < 0.05$, ASR = −2.0). Neither is it used in open-ended activities ($p < 0.01$, ASR = −2.7) nor in those that seek to carry out summative assessment ($p < 0.05$, ASR = −2.1). Examples of this type of activity managed by the teacher would be the use of Genially by children aged 6–9 years to complete multiplication tables (example 18 in Appendix A Table A3) and by students aged 12–16 years when entering a website and taking part in a game to learn irregular verbs (example 17 in Appendix A Table A3).

**Table 7.** Relationship between software resource use and its management: who manages it and how.

| Category | | Relationship between Categories | Relationship | Statistical Data |
|---|---|---|---|---|
| Software (total) | | Access to the information | + | $p < 0.05$, ASR = 2.3 |
| | | Reproduce information | + | $p < 0.05$, ASR = 2.3 |
| | | Closed task | + | $p < 0.05$, ASR = 2.1 |
| | | Assessment of the process (formative) | + | $p < 0.05$, ASR = 2.1 |
| Who manages the information? | Teacher produces | Produce information | − | $p < 0.05$, ASR = −2.7 |
| | | Open task | − | $p < 0.05$, ASR = −2.3 |
| | | Assessment of the process (formative) | − | $p < 0.05$, ASR = −2.0 |
| | Student produces | Reproduce information | − | $p < 0.01$, ASR = −3.0 |
| | | Produce information | + | $p < 0.001$, ASR = 4.4 |
| | | Closed task | − | $p < 0.001$, ASR = −3.3 |
| | | Open task | + | $p < 0.001$, ASR = 3.9 |
| | | Assessment of the process (formative) | + | $p < 0.001$, ASR = 3.5 |

On the contrary, as we saw before, when the software is produced by the student, the activities are open-ended ($p < 0.001$, ASR = 3.9), directed towards the production of information ($p < 0.001$, ASR = 4.4), and their evaluation is more formative ($p < 0.001$, ASR = 3.5). Therefore, when the student manages the resource, it refers to a completely different style of activity, as we can see in the following example in which a teacher instructs the 12–16-year-old students to program through the platform https://scratch.mit.edu/ (accessed on 3 January 2023) (or similar) and then create their own video game (example 20 in Appendix A Table A3).

To sum up, the data show that audiovisuals, text files, software programs, and communication platforms are used in individual activities, primarily aimed at providing students with access to the content delivered and, in many cases, to promote verbal learning. It seems therefore that the resources' uses are more content-centred than student-centred and not very dialogical.

The opposite relationship is also apparent. On the few occasions when students manage resources, assessment is more formative.

Finally, regarding our fourth objective, we found no relevant effects related to teaching characteristics. The use of these resources was considerably homogeneous in all the conditions analysed, which is highly relevant, given the diversity of the educational spaces analysed, and considering the educational levels studied and subjects included. Nevertheless, we found some obvious data, such as that text files were more widely used with 12–16-year-old students ($p < 0.05$, ASR = 2.3) than with children between 6 and 9 years of age ($p < 0.05$, ASR = −3.0), where specifically expository texts were hardly used (ASR = −2.6, $p < 0.05$). We also found a predominance of expressive text use by females over males (ASR = 3.0, $p < 0.01$) which might have indicated a certain tendency of females to promote attitudes. However, the latter was not reflected in the data.

## 4. Discussion

As was proposed in the Introduction, participation in today's digital society requires the development of a set of competences that would be very difficult to learn without the use of digital technologies in education. We also saw that, to achieve these outcomes, these digital resources must be used in formats that facilitate student-centred teaching, through dialogical situations involving multiple voices, acknowledging a diversity of viewpoints, and using multiple codes and languages [3–5]. However, the results we have presented show that teachers' uses of ICT during classroom lockdown fall far from these requirements. As we have seen, teachers used digital resources predominantly to send information or content to their students, mainly in audiovisual format and via text files.

Social media platforms were also used more to transmit information or work instructions in a unidirectional way (generally from the teacher to the students) than to generate spaces for dialogue and interaction. Educational software was hardly used, with these scarce uses mainly focused on repetitive assessments through games such as Kahoot and escape-room. The use of simulations that allow interaction with environments inaccessible from the classroom was practically non-existent [41].

In short, digital resources were used more to transmit information than to engage students in a dialogue with different information or points of view to construct their own views. As we saw in Table 3, a sketch of these tools shows that they were particularly used in closed tasks, in which it was sufficient for students to access the information proposed by the teacher with little or no need to analyse it, let alone organise it. Students' work was oriented towards the reproduction of verbal information and the final product was evaluated more than the process followed to deal with that information. Therefore, our first conclusion is that teaching during the pandemic was fundamentally focused on content. The widespread and necessary use of ICT did not serve to make learning student-centred but was managed by the teacher and primarily aimed at the transmission and evaluation of the content. This pattern is repeated, with some slight variations when we analyse the relationship between the different resources and activities, although these data also show that when activities relinquished control of ICT management to the students to do more than simply return the product to the teachers, teaching was more constructive in its nature. Although less frequent, when such activities occurred, they promoted other ways of learning that were closer to those goals set for ICT integration in the curriculum [1].

The dialogical possibilities of these tools were not exploited. The data show that the most common activities were individual or whole-group activities. There was hardly any student work in small groups, so collaborative or cooperative tasks among students were rare. Therefore, the second conclusion is that ICT during the pandemic also did not serve to move in the direction of more dialogical teaching.

Regarding the use of different codes and languages, the type of activity analysis we have carried out does not inform whether teaching was directed towards multimodal learning based on multiple representations [5]. Although within the text files there is a preference for texts that combine different representational formats (multiple texts) and there is often a combination of audiovisual and textual resources, the emphasis on verbal outcomes seems to indicate that these representations were probably not favoured either. In fact, most audiovisual activities managed by the teacher were aimed at providing students with access to different content. In this sense, the low number of activities aimed at working on students' attitudes is striking, especially when numerous studies [28,42,43] show teachers' concern for their students' wellbeing and emotions during the pandemic. Activities that promoted the learning of procedures are more interesting insofar as they helped in some cases to promote more complex and deeper processing, aimed at the analysis or organisation of data, and not, as was mostly the case, at their mere acquisition or communication, where the students' learning activity once again appeared to refer to the old behaviourist times in which activity was reduced to receiving input and generating an associated output.

Therefore, our results show that the critical incident provoked by the pandemic, which triggered the necessary use of digital technologies to provide continuity in education, led teachers to adapt these resources to more traditional teaching, whose features [8] are confirmed in this study. Undoubtedly, the need to use ICT as the only teaching medium required a great deal of effort from teachers, which may have contributed to this result. Nevertheless, these conclusions are more in line with the results of international studies, mentioned in the Introduction, on the educational use of ICT. These studies show little transformative power and poor learning outcomes [17,18], unlike the experimental works which incorporate more dialogical and student-centred uses and achieve better learning outcomes for students [11]. For the integration of ICT in the classroom to serve to transform teaching and learning practices, it is first necessary to transform teaching conceptions

and practices [44,45], which still seem largely anchored in traditional goals and methods, centred on the transmission of essentially verbal content by the teacher in individual learning spaces.

The data from this study also show that these more traditional practices prevail at all educational levels, in all knowledge domains, and among teachers with different levels of professional experience, as there are hardly any differences in the uses of ICT depending on any of these variables. These fairly homogeneous uses do not seem to depend on these variables, but rather form part of a dominant educational culture whose transformation will require intervention at diverse levels, and also a major effort to promote teacher training in new educational uses of ICT.

Given that some studies show a gap between what teachers say about these uses and what they do in their actual classroom practice [46], it is necessary, on the one hand, for research to differentiate these two components in teaching activity, beliefs, and practices. In this paper, we did not ask teachers about their beliefs but asked them to describe the activities they carried out. We chose this method because asking for a description of activities allowed us at the same time to reach a larger number of teachers than observation would have, and to analyse activities that were closer to those carried out in the classroom. A new system for analysing teaching and learning practices (SATA) was used for this analysis, which we believe is another relevant contribution of this study, as it allows for a more detailed analysis of activities than a closed-ended questionnaire would have provided. However, asking teachers to describe their activities in writing is still a limited methodology for a qualitative analysis of teaching practice. To explore the full spectrum of teachers' explicit conceptions to their classroom practices, a convergence of different methodologies is needed, from experimental or correlational analyses to descriptive studies or case studies.

In future studies, we would like to probe more deeply into an analysis of cases centred on good practices in the use of ICT, understanding as such those that involve a dialogical, student-centred use which integrates different codes or representational modalities. Although most of the tasks sent to us by teachers responded to the profile we have mentioned, and although we did not find significant differences associated with teachers' characteristics, some teachers carried out highly suggestive activities in which they presented open tasks that encouraged their students to collaborate with their classmates and required analysing and organising different information to solve problems, make decisions, or reach their own conclusions. In these tasks the teacher acted more as a guide and assessment formed part of the learning and teaching process itself.

In conclusion, we believe that teacher training, both initial and in-service, should be based on analysing these good practices. Rather than resorting, here too, to training based on verbal explanations to teachers about these good practices, teacher training based on reflection on their own practice is required [27,47]. If teachers have the embodied experience of participating in teacher education in which digital resources are integrated and oriented towards these new uses, only then will they be able to use them accordingly in their teaching practice. Teachers need to realise not only that good integration of digital resources involves designing tasks in which learners manage these resources, but also that this requires learning new teaching roles and functions. It is not about using ICT to channel the teacher's voice but about guiding students to learn to open a dialogue with the multiple voices that can be identified in these environments.

**Author Contributions:** Conceptualization, B.C., M.P.P.E. and J.I.P.; methodology, B.C., M.P.P.E. and J.I.P.; software, J.I.P.; validation, B.C.; formal analysis, B.C.; investigation, B.C., M.P.P.E. and J.I.P.; resources, J.I.P.; data curation, B.C. and J.I.P.; writing—original draft preparation, B.C., M.P.P.E. and J.I.P.; writing—review and editing, B.C., M.P.P.E. and J.I.P.; visualization, B.C.; supervision, B.C., M.P.P.E. and J.I.P.; project administration, J.I.P.; funding acquisition, J.I.P. All authors have read and agreed to the published version of the manuscript.

**Funding:** This research was funded by the Spanish Ministry of Science & Innovation [PID2020-114177RB-I00].

**Institutional Review Board Statement:** The study was conducted in accordance with the Declaration of Helsinki, and approved by the Ethics Committee of University Autonomous of Madrid (CEI-104-2012, 11 February 2022).

**Informed Consent Statement:** Informed consent was obtained from all subjects involved in the study.

**Data Availability Statement:** The datasets generated during and/or analysed during the current study are available from the corresponding author on reasonable request.

**Acknowledgments:** We wish to thank Ricardo Olmos for sharing their statistical knowledge with us. Finally, we would also thank Krystyna Sleziak for her support in the preparation of the English version of this paper.

**Conflicts of Interest:** The authors declare no potential conflicts of interest with respect to the research, authorship, and/or publication of this article.

## Appendix A

**Table A1.** Teachers' variables.

| Variable | Categories | Category N * |
|---|---|---|
| Gender | Men | 77 |
| | Women | 190 |
| | Non-binary ** | 2 |
| Teaching experience | 5 years or fewer | 80 |
| | From 6 to 15 years | 72 |
| | From 16 to 25 years | 75 |
| | 26 years or more | 42 |
| Educational level | 1st, 2nd or 3rd stage of primary education (6–9 years old) | 60 |
| | 4th, 5th or 6th of primary education (9–12 years old) | 68 |
| | Compulsory secondary education (12–16 years old) | 114 |
| | Non-compulsory secondary education (16–18 years old) | 27 |
| Primary curriculum subjects | Generalists | 69 |
| | Specialists | 55 |
| Secondary curriculum subjects | Spanish language | 25 |
| | Mathematics | 12 |
| | Social sciences | 15 |
| | Natural sciences | 26 |
| | Foreign language | 19 |
| | Technology | 14 |
| | Others *** | 29 |
| Type of centre | Public | 231 |
| | Non-public | 40 |
| Previous ICT use | Never | 63 |
| | Several days per month | 125 |
| | Several days per week | 48 |
| | Every day | 33 |

* The number of participants may vary slightly from one variable to another due to some teachers not providing information on that variable. ** In gender analysis non-binary participants were not considered because it was a very small sample. *** The category "others" referred to specialities with a very small number of teachers.

**Table A2.** Examples of activities: type of resources used.

| Digital Resources Used | | Example According to the Digital Resource Used |
|---|---|---|
| Text files | Informative | Search the internet for a technological invention that is scheduled from the year 2025 to 2030. [ . . . ] look for information (as much as possible) about this technological invention that has not been developed in Spain (1) |
| | Questionnaire-based | Review of some musical language elements covered during the course. Review of vocabulary to express how instruments are played. Questionnaire on the theoretical content of the course, which it can consult in a free copyright book in pdf or photocopies (2) |
| | Expository | Students sent me an argumentative text describing the area of improvement in their school and the justification for their decision (3) |
| | Expressive | Take sound samples of the sounds of confinement, record their origin, and write or compose a poem or song about it, sensations, etc. (4) |
| | Multiple | Search for information, select it and do a short piece consisting of an explanation with a drawing or image, a question, and its answer (5) |
| Communication platforms | Social media platforms | The English classes are developed with material that is shared by me in the subject's blog. This material includes both written texts and audiovisual material (6) |
| | Email | Electronic email (means of communication for sending things, queries, etc.) (7) |
| | Video call | Videoconferences with CISCO WEBEX (8) |

**Table A3.** Examples of activities: resource management.

| Digital Resources Used | Example According to Digital Resource Management | | | |
|---|---|---|---|---|
| | Teacher | | Student | |
| | Selection | Production | Selection | Production |
| Audiovisual | Viewing of a video about an initiative in Madrid entitled: "Madrid, te comería a besos" (Madrid, I'd cover you in kisses). | To host in the Educamadrid Media Library links to totally original audiovisual montages of their own creation. Students must access them and answer the questions posed to them and create presentations and/or carry out blankquizz questionnaires (10) | Searching the internet for information, reading the information in the books or documents of each region and even watching some video reporting (11) | Creative writing workshop with book covers where the students have recorded themselves doing it (12) |
| Text files | The reading of a short extension of a book to make different contributions to its main ideas. On the premise that everyone has to participate (13) | It is prepared "swat notes" where the information previously explained is written appealingly (14) | Recording of a video reading their favourite book (15) | Students should be creative and put into practice what they had learned about the argumentative text. It should be written a text of no more than 100 words in which is recounted the encounter with a character with appropriateness, coherence, and cohesion (16) |
| Software | Going into a website where students used a game to learn the irregular verbs (17) | Going into a website where students used a game to learn the irregular verbs (17) | Not applicable (19) | Not applicable (19) |

**Table A4.** Examples of activities: learning outcomes and processes.

| Learning Outcomes | Example According to the Process Activated | |
|---|---|---|
| | **Reproductive** | **Constructive** |
| Verbal | Watch an EDpuzzle video story and answer questions on a sheet of paper they then had to hand in (21) | Select a work of art and its artist. Describe it according to the criteria of colour, and composition seen in plastic and visual arts. Then reflect on the political, economic, and social situation during the time of its creation (22) |
| Procedural | Take a picture of the notebook and rename the file "homework of such and such person_1A.jpg" Attach the picture to the message. Send it to your teacher and copy (CC) my email. In "subject" write something specific. "English homework". Such and such a person (1 A)". Write the message including a greeting, polite sentence, clear and concise message, farewell, and signature (giving them an example). Check the spelling. Optionally, Students can convert converting the image to PDF format (23) | Split into groups of 5 or 6, divide up tasks, look for and summarise information on a chosen theme (a food web, the Nazis and the holocaust, coronavirus, addictions). Outline the information using PowerPoint (24) |
| Attitudinal | The objective was that students would do a bit of exercise but, above all, that they have a good time and learn to interpret the song and eliminate stage fright of individual interpretation (25) | Students should reflect on the conflict resolutions in daily life, taking into account the different attitudes which could be adopted towards a problem, and relate them to their behaviour at different times. (26) |

**Table A5.** Examples of activities: characteristics of activities.

| Characteristics of Activities | | Examples According to the Characteristics of Activities |
|---|---|---|
| Information processing | Acquisition | Through ClassDojo it is given a pdf document where the student will find "notes" on new learning. External links are included (YouTube, web documents, etc.). The course ends with a knowledge acquisition test (27) |
| | Interpretation | Work on the recognition of emotions through their reference teachers. For this objective, it is created an interactive book for each of the basic emotions in which students have to identify their reference teacher, observe the emotion reflected in their facial gesture and associate it with its corresponding pictogram (28) |
| | Analysis | Make a video tutorial of how they would determine an unknown variable in a real cinematic case made in their homes (it was during total lockdown) (29) |
| | Organisation | Research on a social or natural topic in a group [ . . . ]. Analysis of the information read. Students should schematise it and make the presentation (30) |
| | Communication | Activity for the second year of Secondary Education. The activity aims to motivate students to collect information on a topic of personal interest but related to the world of young people and then be able to present it in a graph and explain the information obtained in English (31) |
| Information management | Access | Viewing of a video recorded by the teacher and the conversation assistant that the students had the previous year (32) |
| | Search | Natural science work on nutrition and health. After finishing the topic, they had to look for a curiosity related to the content of the topic, make up a question and its answer (33) |
| Production/ Reproduction of the information | Reproduce | Watch a documentary on YouTube and answer several questions (34) |
| | Produce | This is a small project with the following steps 1—Analysis of the reality of the school and detection of a possible aspect for improvement (initial metacognition) 2—Conducting a survey about this aspect of improvement of the school 3—Data collection with the answers given to the survey by colleagues 4—Analysis of results 5—Presentation of the conclusions in a digital portfolio and reflection of the work process (final metacognition) 6—Extension of the digital portfolio by selecting other work from the course with its corresponding reflection (35) |

<p style="text-align:center">Table A5. *Cont.*</p>

| Characteristics of Activities | | Examples According to the Characteristics of Activities |
|---|---|---|
| Task | Closed | Students accessed to Genially and worked day by day, unlocking the next episodes. They also self-corrected the exercises and performed self-assessments. (36) |
| | Open | A research project related to the social studies area. It is called "travelling around the world". It consists of choosing a place we would like to visit. Each week it is given guidelines about the project. Students have to research customs, currency, language, etc. Besides, they should make a budget of approximate expenses it would entail the trip (37) |
| Evaluation | Product/summative | At the end of the course, there is a knowledge acquisition test. This is a self–corrected test designed with Liveworksheets (38) |
| | Process/formative | Using different rubrics for each part of the process and with direct feedback from me, both in the shared documents and by email. In addition, several questions for co-evaluation of the peer work were included in all the surveys (39) |
| Social organisation | Individual | Watching a short film and answering questions that related what happened in the film to their daily lives (40) |
| | Class group | From time to time, it is made video calls to hold competitions or tournaments on the content which is worked on [—]. These tournaments had rounds of questions, and the pupils showed their answers with a small blackboard on which they wrote the solution. Students, when they are right, get points in the ClassDojo. Students have been highly motivated and eager to work and learn with each question they have been asked (41) |
| | Group work | Students are asked in groups of three to solve a problem related to curriculum content. The problem include several questions or issues to investigate (42) |

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
