# Peer review of "The Use of Digital Resources in Teaching during the Pandemic: What Type of Learning Have They Promoted?"

_education, doi:10.3390/educsci13010058_

Round 1

Reviewer 1 Report

I understand that this paper acknowledges that the digital resources analysed mainly focused on traditional teaching and proposes reinforcing more student-centred activities. However, after reading this, I feel that the authors do not emphasise the context in which the teachers were in general. I think it is imperative to highlight the great work that all teachers have done. Although during the pandemic, they did not reach an advanced level in dialogic and student-centred teaching, it is essential to reflect on this experience so that we can reach this new goal in the near future.

I would recommend it for publication in the journal “Education Sciences” after major revisions. I hope the authors take these corrections in good faith, as I intend their work to be published with the best possible quality. From now on, I will use this format: P=page, L=line, example: P2 L33.

·      Keywords: I recommend adding “Digital resources”, “ICT”, and “COVID-19” as keywords.

·      Check the grammar, spelling, and syntax of the entire document.

·      Review the English of the document in detail. Specially check the use of articles and prepositions in sentences. Additionally, check British English, for instance: In table 1. “social organization”. The same is in Table A2. For example, check Table 1: Explanation of software.

·      Unify the format of all tables. Check all table captions.

·      Be careful with the use of repeated words within the same paragraph and throughout the document. For example, “in fact” (P14 L482 and P14 L485).

·      P3 L137a. At what point in the pandemic was this study conducted? I think this information is essential because, at the beginning of the contingency (March 2020), it was a very complicated time for everyone, where teachers did everything they could to be able to finish the semester. This pandemic has brought new knowledge and skills to teachers, especially in technology. Therefore, if this study was conducted long after the onset of the contingency, it is arguable how much progress was made in this area.

·      P3 L137b. The format in which the activities were done is not explicit anywhere in the paper. Was it online, by TV, or how? This should be clarified and reinforced throughout the paper.

·      P3 L137c. Was information collected on whether the teachers were from public or private schools? I think this is a crucial aspect as to what resources the institutions provided to them during contingency.

·      P3 L141 / P8 L256 / P8 L258. P8 L263. P8 L265. P8 L301. P8 L303. (And so on). Past tense should be used.

·      P2 L93. Some examples can be added, which normally are platforms from different editorials such as Pearson, CENGAGE, McGraw-Hill, etc.

·      P3 L102. Take care of expressions such as “Put differently…”. It can be rewritten as “In other words…”

·      P3 L120-122. This sentence needs to be referenced.

·      P3 L128. Consider rephrasing: “…sources have been used in teaching during the pandemic…” by “…sources have been used in teaching in Primary and Secondary Education during the pandemic…”

·      P3 L131. The expression “to promote” should be deleted, or the sentence should be rewritten.

·      P4 L176. Do not use bold type in "categorisation".

·      Table 1. Check punctuation and text in the table (table + caption).

·      Table 1. (And so on). “Questionnaire” should be “Questionnaire-based”.

·      Table 1. Category “Mail” should be “email”. This word must be unified in the article.

·      P6 L202. Table 1 should be placed at the end of section 2.3.

·      P6 L209. Lines 209-201 should be rewritten.

·      Figure 1. y-Axis should be “Number of digital resources” more than “Frequency of use”.

·      P8 L270. P8 L272. Table 3. (And so on). Use the minus sign (−), not (-).

·      P8 L281. Modify the table caption for better understanding. The same is in Table 6 caption.

·      Table 3. There is a number “283” below “Category” that must be removed. Check all tables and figures.

·      P9 L300. Lines 300-305 are difficult to understand.

·      P9 L304. Example 18 is not representative of the explanation exposed.

·      Table 1. The definition of “Multiple texts” says that they are texts which present a combination of two or more codes (multimodality). However, in P9 L304, multiple texts are referred to as several texts in the same activity. Define this concept more clearly and unify it.

·      P10 L324 (And so on). “p” should be in italics.

·      P11 L363. Check font type.

·      Table 6. “Open task” should be capitalised.

·      P11 L378. “Audio-visual” should be written as “Audiovisual”, and it must be unified all over the article.

·      P12 L382. Lines 382-383 should be revised and rewritten.

·      P12 L391. No discussion of gender has been made until this point. I assume this is about the gender of teachers, although the previous sentence refers to students. Rewrite.

·      P13 L430. I believe that the timing of the study should be contextualised after the second conclusion, as it surely influenced the failure to achieve more dialogic teaching.

·      Table A1. The asterisk in “Others* **” should be checked. Also, review grammar in the notes.

·      Table A2. “Who manages and how” is very difficult to read.

·      Table A2. “Educamdrid Media Library”. Is it well-written?

·      Table A2. “PC” must be capitalised.

·      Table A2. All examples should be revised and rewritten to facilitate reading, unifying the format (for example, indicating the objective of the activity). Many of them were very difficult to read and understand the topic.

·      Table A2. Task >>> Closed. A form with questions about the interview is not representative of this example (at least, not how it is expressed). The definition of closed task is: “Tasks with well-defined procedures where the answers may be classed as correct or incorrect”.

·      Bibliography must be revised. Check the size and type of font, use of dash line (“−“, not “-“), and initial spacing in reference 11.

Although the reader may find the article easy and enjoyable, I encourage the authors to revise the manuscript as recommended above for publication. Congratulations on the work done!

Author Response

We would like to thank all reviewers and the editor for their careful reading, reflections, and contributions. In our opinion, all these contributions contribute to improving the text for which we are very grateful. In the following lines we will indicate the changes we have made, all of them marked through the control of changes in the manuscript. We will also indicate the few changes we have not made and explain the reasons for not having done so.

Reviewer 1

Writing problems

Q1. Check the grammar, spelling, and syntax of the entire document.

We thank the reviewer for the detailed review of the language used in the manuscript. The words or expressions you have pointed out to us have been modified, but also many others. For this reason, we do not indicate the exact places of the changes, which are visible by means of control of changes. On the other hand, a native reviewer has monitored and controlled the entire document.

Format manuscript

Q2. Problems related to the manuscript format

We have modified all the reviewer’s suggestions related to the manuscript format. For example, table captions, and errors in the format of tables have been unified. Errors in bold and the font type of the entire manuscript have also been corrected. In addition, "-" has been replaced by "−" to indicate the negative score and added in italics "p" in the p-values. Finally, "audio-visual" has been replaced with "audiovisual".

Q3. Figure 1. y-Axis should be “Number of digital resources” more than “Frequency of use”.

We thank the reviewer for his contribution; however, we have decided not to modify the expression "Frequency of use" from the Y-axis to "number of digital resources". We understand that frequency refers to the number of times an event is repeated. On the other hand, the term "number of resources" can be understood as the total number of these resources. In addition, Figures 2 and 3 and Table 2 carry the word frequency in their titles with the same sense as Figure 1.

Abstract and keywords

Q4. I recommend adding “Digital resources”, “ICT”, and “COVID-19” in Keywords.

We have added the keywords "Digital resources", "ICT" and "COVID-19" and we have eliminated the keywords, "Distance education and online learning". Finally, we have replaced "Information literacy" with "Information processing" and merged "Elementary and Secondary education".

Introduction

Q5. Some examples can be added, which normally are platforms from different editorials such as Pearson, CENGAGE, McGraw-Hill, etc.

We have added some examples of platforms (lines 97-98).

Q6. This sentence needs to be referenced: “Other studies have found an inverse relationship between teachers' age or years of teaching experience and the way in which digital resources are accepted, while others do not confirm these differences”.

We have introduced the citations necessary (lines 125-126).

Q7. Table A2. All examples should be revised and rewritten to facilitate reading, unifying the format (for example, indicating the objective of the activity). Many of them were very difficult to read and understand the topic.

The examples of activities refer to full examples of teachers' descriptions. We believe that it is not pertinent to change the words of the participants. However, we have included some changes to improve the grammar and syntaxis on the table. 

Method

Q8. At what point in the pandemic was this study conducted? I think this information is essential because, at the beginning of the contingency (March 2020), it was a very complicated time for everyone, where teachers did everything they could to be able to finish the semester. This pandemic has brought new knowledge and skills to teachers, especially in technology. Therefore, if this study was conducted long after the onset of the contingency, it is arguable how much progress was made in this area.

We have added at the beginning of the method when the data was collected (lines 147). In addition, we have included both in the introduction (lines 85-86) and the discussion (lines 458-459) the difficulty posed by the moment of confinement and the effort made by teachers to teach during this period.

Q9. The format in which the activities were done is not explicit anywhere in the paper. Was it online, by TV, or how? This should be clarified and reinforced throughout the paper.

We are not quite sure what the reviewer wants to tell us about this indication. We believe that the Category System delves into what the characteristics of the activity were and specifically also considers the media in which the activities were carried out. On the other hand, the different Spanish regions (responsible for compulsory education) have their own educational platforms (for example, EducaMadrid) to which the different ICT resources are uploaded. If the reviewer refers to this last aspect, we can include it in the text

Q10. Was information collected on whether the teachers were from public or private schools? I think this is a crucial aspect as to what resources the institutions provided to them during contingency.

We have added in Appendix A1 the number of teachers in public and non-public schools.  In addition, we have included this variable in line 156. We have not introduced the word private schools because in Spain there is a high number of privately run schools but with public funding. They are the “centros concertados” (concerted centres). We have introduced in the table the two types of centres. However, this variable, like other independent variables, did not affect the type of activities that teachers sent. For this reason, it is not introduced either in the results or in the conclusions.

Q11. Example 18 is not representative of the explanation exposed.

That example has been modified (line 308).

Q12. Table 1. The definition of “Multiple texts” says that they are texts which present a combination of two or more codes (multimodality). However, in P9 L304, multiple texts are referred to as several texts in the same activity. Define this concept more clearly and unify it.

The reviewer is right in this assessment. The category multiple texts refers to multimodality. We have removed the explanation between parenthesis in line 297.

Appendices

Q13. Table A2. Task >>> Closed. A form with questions about the interview is not representative of this example (at least, not how it is expressed). The definition of “Closed task” is: “Tasks with well-defined procedures where the answers may be classed as correct or incorrect”.

We have changed this example in Table A2d.

Reviewer 2 Report

This article presents an investigation into what type of digital resources were used during covid, compared with their frequency of use.
The work is clear, relevant to the field and presented in a well-structured manner. The attempt to investigate how the type of resource is related to the type of learning that was intended to be used is interesting.
Some references are a little outdated, but I think they are still relevant. The 20% self-citation could be lowered. Part of the methodology refers to articles by the authors themselves which are, for obvious reasons, not quoted but do not help the understanding of the paper. Lines 154-158 could be expanded with some further explanation of the category systems and their focus.
It is also unclear how McNemar's test for objective 2 is used; how is the table for the analysis constructed? Which variables are considered?
The conclusions are consistent with the evidence and arguments presented. Very long tables do not help the reading of the work. It would perhaps also be useful to have a summary table of the test results, given their number and their dispersion in the text.
There is also a formatting error in the text, on line 363.

Author Response

We would like to thank all reviewers and the editor for their careful reading, reflections, and contributions. In our opinion, all these contributions contribute to improving the text for which we are very grateful. In the following lines we will indicate the changes we have made, all of them marked through the control of changes in the manuscript. We will also indicate the few changes we have not made and explain the reasons for not having done so.

Reviewer 2

Format manuscript

Q1. There is also a formatting error in the text (line 363).

This mistake has been corrected.

Method

Q2. Lines 154-158 could be expanded with some further explanation of the category systems and their focus.

Section 2.3 specifies the Category System used. In addition, in Tables A2 of the annexes, we include an example of an activity for each of the SATA categories.

Q3. It is also unclear how McNemar's test for objective 2 is used; how is the table for the analysis constructed? Which variables are considered?

We have added a paragraph explaining how McNemar's test was used (lines 213-217).

Discussion

Q4. Very long tables do not help the reading of the work. It would perhaps also be useful to have a summary table of the test results, given their number and their dispersion in the text.

We thank the reviewer for his contribution which we believe helps to make the article more understandable. However, instead of making a summary table, we have chosen to subdivide tables 1 and A2 in the appendix to make them more understandable. In accordance with the complexity and variety of results, the resulting summary table was very long and complex and we did not find it useful to include it

References

Q5. Some references are a little outdated, but I think they are still relevant.

We agree with the reviewer that there are some old references, but we also agree that they are relevant For that reason we have kept them. We believe that they complement well to other more current references.

Q6. The 20% self-citation could be lowered.

We have removed some of the self-citation. Finally, we have 7 references from the authors out of a total of 47, that is, less than 15% of the total references, which are essential for the justification of this work.

Round 2

Reviewer 1 Report

The modifications made to the document have made it much easier to read. I am sharing new suggestions for the article to be of the highest possible quality. I made some of them last time, but they may have been overlooked. From now on, I will use this format: P=page, L=line, example: P2 L33.

·      Keywords: Keywords must be separated by semicolons. COVID in COVID-19 must be capitalized.

·      P1 26. In “21st century”, do not set the letters denoting ordinals in superscript.

·      P4 L163. “Table” should be in the plural (see Tables 1a and 1b). Same in P6 L213 (and capitalized).

·      P4 L186. Add “by” in “…were managed by the teacher or by the students…”.

·      P5 L205. Check periods in the phrase “…in Appendix A (2a, 2b, 2c, 2d)”.

·      Table 1a. “Teacher” must be capitalized.

·      Table 1b. “Substantially repeat the information” should be rephrased to “Substantial repetition of the information.”

·      Table 1b. Check the caption.

·      Table 1b. There are 3 "+" signs at the end of the table (in Evaluation and in Social organisation) without any explanation.

·      P7 L222. Check the symbol of chi-square. Same in P7 L226.

·      P7 L241. In “p < 0.001” spaces must be placed between the "<" signs. Same in P7 L246.

·      P7 L244. Check capitalization in this phrase (The).

·      P8 L267. The format for writing the appendix should be unified; for example: "Appendix A.2c" (no period in the end). Same in P8 L270.

·      P9 L301-307. This sentence can be separated into 3 shorter ones for better readability.

·      Figure 3. The note is hidden by the figure.

·      P12 L387. This sentence should be part of the previous paragraph.

·      Table A.2a. Modify “Questionnaire” by “Questionnaire-based”. The ellipses are expressed as “…”. “Etcetera” must be written as “etc.” Capitalization of “Resources” is unnecessary.

·      Table A.2b. Modify “Audio-visual” by “Audiovisual”.

·      Tables A.2a-d. The new layout of this table is much more readable right now. However, all entries must have the same format to explain how the resource has been used. I do not recommend writing the phrases with prefixes such as “I” or “They”. For example, in Table A.2a, I suggest you rewrite the “Expository” and “Social media platforms” entries.

·      References. Avoid bold type in commas and periods (entries 2, 3, 4, and 5).

·      Ref. 14. Check space in pages “531- 549”.

·      The bibliography must be revised. Check the use of dash lines (“−“, not “-“) in all cases.

Author Response

We would like to thank reviewer 1 and the editor for their suggestion. All these contributions, again, have contributed to improving this manuscript. In the following lines, we will indicate the changes we have made, all of them marked through the control of changes in the manuscript. We will also indicate the few changes we have not made and explain the reasons for not having done so.

Reviewer 1

Q1. Keywords: Keywords must be separated by semicolons. COVID in COVID-19 must be capitalized. 

We have added the semicolons (line 19).

Q2. P1 26. In “21st century”, do not set the letters denoting ordinals in superscript. 

We have eliminated the superscript (line 26).

Q3. P4 L163. “Table” should be in the plural (see Tables 1a and 1b). Same in P6 L213 (and capitalized). 

We have added the plural in tables 1a and 1b (lines 164, 213)

Q4. P4 L186. Add “by” in “…were managed by the teacher or by the students…”. 

We have added by in that sentence (line 186) 

Q5. P5 L205. Check periods in the phrase “…in Appendix A (2a, 2b, 2c, 2d)”. 

We have eliminated the periods in that sentence (line 205).

Q6. Table 1a. “Teacher” must be capitalized. 

We have capitalized “Teacher” in table 1a.

Q7. Table 1b. “Substantially repeat the information” should be rephrased to “Substantial repetition of the information.” 

We have included that modification.

Q8. Table 1b. Check the caption. 

We have modified the caption in Table 1b.

Q9. Table 1b. There are 3 "+" signs at the end of the table (in Evaluation and in Social organisation) without any explanation. 

We have eliminated the “+” in those cases without any explanation.

Q10. P7 L222. Check the symbol of chi-square. Same in P7 L226. 

We have corrected the chi-square symbol (lines 222, 226)

Q11. P7 L241. In “p < 0.001” spaces must be placed between the "<" signs. Same in P7 L246. 

We have included those modifications (lines 241,246)

Q12. P7 L244. Check capitalization in this phrase (The). 

We have corrected the capitalization.

Q13. P8 L267. The format for writing the appendix should be unified; for example: "Appendix A.2c" (no period in the end). Same in P8 L270. 

We have unified the format of the titles included in the appendices section (lines 267,270)

Q14. P9 L301-307. This sentence can be separated into 3 shorter ones for better readability. 

We have rewritten this sentence (lines 301-307).

Q15. Figure 3. The note is hidden by the figure. 

We have moved the note in Figure 3 toward the correct place.

Q16. P12 L387. This sentence should be part of the previous paragraph.

That sentence are summing up the idea of the section. We think it is more recommendable to keep the sentence in the same place (Iines 385-389).

Q17. Table A.2a. Modify “Questionnaire” by “Questionnaire-based”. The ellipses are expressed as “…”. “Etcetera” must be written as “etc.” Capitalization of “Resources” is unnecessary. 

We have replaced “questionnaire” with “questionnaire-based” (Table A.2a) and “etc” with “etc.” (lines 416, Table A.2a, A.2d). We have eliminated the capitalization of resources (lines 178-180) and the ellipses which were unnecessary.

Q18. Table A.2b. Modify “Audio-visual” by “Audiovisual”. 

We have included this modification (Table A.2b.).

Q19. Tables A.2a-d. The new layout of this table is much more readable right now. However, all entries must have the same format to explain how the resource has been used. I do not recommend writing the phrases with prefixes such as “I” or “They”. For example, in Table A.2a, I suggest you rewrite the “Expository” and “Social media platforms” entries.

We have rewritten some entries according to the reviewer’s recommendations. However, we have tried to keep the “words” of the teachers (Table A.2a, b, c and d).

Q20. References. Avoid bold type in commas and periods (entries 2, 3, 4, and 5).

We have included that modification.

Q21. Ref. 14. Check space in pages “531- 549”.

We have eliminated the space in the manuscript.

Q22. The bibliography must be revised. Check the use of dash lines (“−“, not “-“) in all cases.

We have checked the biography according to the rules of Education Science. We have replaced "-" with “−“ in the manuscript.